# Melatonin Supplementation Alleviates Impaired Spatial Memory by Influencing Aβ_1-42_ Metabolism via γ-Secretase in the icvAβ_1-42_ Rat Model with Pinealectomy

**DOI:** 10.3390/ijms251910294

**Published:** 2024-09-24

**Authors:** Irina Georgieva, Jana Tchekalarova, Zlatina Nenchovska, Lidia Kortenska, Rumiana Tzoneva

**Affiliations:** 1Institute of Biophysics and Biomedical Engineering, Bulgarian Academy of Sciences, Acad. G. Bonchev Street, Block 21, 1113 Sofia, Bulgaria; igeorgieva@biomed.bas.bg; 2Institute of Neurobiology, Bulgarian Academy of Sciences, Acad. G. Bonchev Street, Block 23, 1113 Sofia, Bulgaria; janetchekalarova@gmail.com (J.T.); zuzania@abv.bg (Z.N.); lkortenska@abv.bg (L.K.)

**Keywords:** icvAβ_1-42_, β-amyloid, γ-secretase, Alzheimer rat model, pinealectomy, spatial memory, melatonin treatment

## Abstract

In the search for Alzheimer’s disease (AD) therapies, most animal models focus on familial AD, which accounts for a small fraction of cases. The majority of AD cases arise from stress factors, such as oxidative stress, leading to neurological changes (sporadic AD). Early in AD progression, dysfunction in γ-secretase causes the formation of insoluble Aβ_1-42_ peptides, which aggregate into senile plaques, triggering neurodegeneration, cognitive decline, and circadian rhythm disturbances. To better model sporadic AD, we used a new AD rat model induced by intracerebroventricular administration of Aβ_1-42_ oligomers (icvAβ_1-42_) combined with melatonin deficiency via pinealectomy (pin). We validated this model by assessing spatial memory using the radial arm maze test and measuring Aβ_1-42_ and γ-secretase levels in the frontal cortex and hippocampus with ELISA. The icvAβ_1-42_ + pin model experienced impaired spatial memory and increased Aβ_1-42_ and γ-secretase levels in the frontal cortex and hippocampus, effects not seen with either icvAβ_1-42_ or the pin alone. Chronic melatonin treatment reversed memory deficits and reduced Aβ_1-42_ and γ-secretase levels in both structures. Our findings suggest that our icvAβ_1-42_ + pin model is extremely valuable for future AD research.

## 1. Introduction

Alzheimer’s disease (AD) is defined as the main cause of dementia and affects approximately 45 million people worldwide [1]. In most cases, AD progression is initiated in the elderly, at an average age of 65 years. One of the most widely discussed hypothetical models of AD pathogenesis involves the accumulation of different types of Aβ and upstream plaques in the brain, which precede the spread of tau-mediated neurofibrillary tangles, synaptic dysfunction, and neuronal loss. Moreover, amyloid plaques cause alterations in membrane permeability and an increase in oxidative stress, eventually triggering AD-associated clinical manifestations such as memory loss, cognitive dysfunction, and personality changes [2,3,4]. It has been convincingly demonstrated that Aβ increases the rate of neurofibrillary tangle formation in mice, that also express mutant tau protein [5,6], placing Aβ pathology firmly ahead of tangle accumulation in the hierarchy of disease progression [7]. In addition, Aβ accumulation in AD patients causes changes in their circadian rhythms and increased oxidative stress in neuronal tissue before the onset of any cognitive disturbances [8], supporting the role of Aβ in AD progression.

Amyloid β is a 4 kDa fragment of the larger amyloid precursor protein (APP), which is produced primarily by brain neurons, vascular and blood cells, and to a lesser extent by astrocytes [1]. APP trafficking is an essential factor in APP metabolism and is driven by three secretases: α-, β-, and γ-secretases. One possible explanation for AD is the dysregulation of the secretases and the consequent changes in the processing of APP [9,10]. For instance, APP matures in the endoplasmic reticulum and Golgi apparatus, after which it can be translocated to the cell surface or it can enter the lysosomal pathway and undergo proteolytic degradation [11]. When it is located at the plasma membrane, the APP is normally processed by α-secretase to form a non-amyloidogenic soluble peptide that is secreted into the extracellular space and is involved in intercellular communication, synaptic plasticity, neurogenesis, and long-term potentiation. Alternatively, the APP goes through the amyloidogenic pathway, where β- and γ-secretases sequentially cleave it into two types of protein fragments: a secreted aggregation-prone fragment (42 amino acids long, Aβ_1-42_) and a membrane-associated fragment, the former being the main culprit in AD.

γ-secretase (GS), as a player in APP processing, is primarily responsible for the formation of toxic Aβ species [12]. The enzyme consists of four different proteins: presenilin (PS), nicastrin (Nct), anterior pharynx-defective 1 (Aph-1), and presenilin-enhancer 2 (Pen-2), present in a 1:1:1:1 stoichiometry, all of which need to be expressed together for maximal GS activity [13]. Under normal circumstances, GS cleavage produces shorter non-amyloidogenic peptides, Aβ_1-37_, Aβ_1-38_, and Aβ_1-40_, which play physiological roles in synaptic plasticity, neurogenesis, and long-term potentiation [14]. Unfortunately, GS tends to be “messy” and accidentally produces oligomeric and insoluble Aβ (Aβ_1-42_) peptides, which are neurotoxic because they aggregate, form senile plaques, and initiate a sequence of changes leading to neuroinflammation and ultimately to AD [3,7]. A study using transgenic mice showed that amyloid deposition is driven almost entirely by Aβ_1-42_ and not by Aβ_1-40_ [15]. Furthermore, mutations in PS reduce the catalytic activity of GS and contribute to an increased Aβ_1-42_:Aβ_1-40_ ratio in familial AD [16,17]. As a result, the proportion of slightly longer and more hydrophobic and fibrillogenic forms of Aβ, particularly Aβ_1-42_, increases, making them the major species deposited in the AD brain [18].

Disruption of the circadian rhythm in AD patients is among the most commonly observed cognitive changes [19,20], but has not been thoroughly investigated. In fact, AD causes significant alterations in the pineal gland (responsible for melatonin synthesis), such as calcification and reduction in size [21]. As a result, over 60% of AD patients suffer from sleep disturbances, which worsen cognition and exacerbate neurodegeneration [22]. Along with pineal gland dysfunction, AD patients often lose intrinsically photosensitive retinal ganglion cells, crucial for regulating melatonin secretion [23]. This contributes to “sundowning”, marked by confusion and agitation in the late afternoon, along with sleep disturbances and insomnia [24]. Notably, Whittaker et al., 2023 [8] introduced fasting as a way to improve the circadian rhythm of mice with familial AD, which enhanced the removal of Aβ_1-42_ and consequently led to reduced neuroinflammation. Moreover, melatonin supplementation has consistently been found to improve cognitive deficits, support learning, and lower Aβ_1-42_ levels by both limiting its production and enhancing its clearance, as demonstrated in a recent meta-analysis [25].

Previously, we introduced a new rat model of AD with induced melatonin deficiency through pinealectomy (pin) and intracerebroventricular (icv) infusion of Aβ_1-42_ oligomer [26]. We demonstrated that the combination of pin and icvAβ_1-42_ provoked significant changes in anxiety and oxidative stress in the animals compared to either pin or icvAβ_1-42_ groups alone. Moreover, exogenous melatonin treatment of the rat AD model attenuated the observed behavioral changes and oxidative stress elevation. In the present study, we aimed to validate further our established animal model of AD (icvAβ_1-42_ + pin) and confirm that it worsens spatial memory in rats and enhances Aβ_1-42_ production via increased γ-secretase levels in the frontal cortex and hippocampus. We also tested whether melatonin supplementation could correct these alterations.

## 2. Results

### 2.1. Chronic Melatonin Treatment Mitigated the Impaired Spatial Memory in the Aβ_1-42_ Rat Model with Pinealectomy

A radial arm maze (RAM) test was used to measure the working memory of all eight animal groups. By measuring the animals’ working memory we can assess their ability to temporarily store information, i.e., to enter each of the eight arms of the maze just once. A re-entry into an arm is called a working memory error (WME). We used a two-way repeated ANOVA test to gain insight into the effect of Aβ_1-42_ and pinealectomy on the working memory capacity of our rat models. The test revealed a main Time (number of sessions) [F4,199 = 17.541, *p* < 0.001] and a significant time × group interaction [F32,199 = 6.420, *p* = 0.004] for the number of working memory errors (WMEs) across the five sessions. Notably, most of the melatonin-treated groups demonstrated an efficient learning capacity to perform the memory task, as revealed by the post hoc test: C-sham-veh group (fourth vs. first session, *p* = 0.007, fifth vs. first session, *p* < 0.001), for the C-sham-mel group (fifth vs. first session, *p* < 0.001), for the C-pin-mel group (fifth vs. first session, *p* = 0.004), and for the Aβ-pin-mel group (fifth vs. first session, *p* < 0.001) (Figure 1A). In contrast, there was no significant difference in the WMEs over the five sessions for the C-pin-veh, Aβ-sham-veh and Aβ-pin-veh groups (*p* > 0.05), suggesting that the animals are unable to learn the spatial memory task in the RAM test. In addition, these groups had significantly more WMEs during the last, fifth, trial of the RAM test (*p* < 0.001, C-pin-veh vs. C-sham-veh group; *p* = 0.026, Aβ-sham-veh vs. C-sham-veh; *p* < 0.01, Aβ-pin-veh vs. C-sham-veh). Melatonin supplementation corrected the impaired memory response measured in the fifth session of the RAM test (*p* < 0.001, C-pin-mel vs. C-pin-veh group; *p* = 0.005, Aβ-pin-mel vs. Aβ-pin-veh group).

Furthermore, the two-way ANOVA test showed the main effect of treatment (Aβ_1-42_ and/or pinealecotmy) [F1,58 = 3.174, *p* = 0.032] and drug (melatonin) [F1,58 = 10.249, *p* = 0.002] on the average WME. The post hoc test indicated that concurrent Aβ_1-42_ infusion and removal of the pineal gland increased the average number of WMEs (*p* = 0.018, C-pin-veh vs. C-sham-veh; *p* = 0.048, Aβ-sham-veh vs. C-sham-veh; *p* = 0.05, Aβ-pin-veh vs. C-sham-veh). At the same time, melatonin treatment alleviated this response only in the Aβ-pin-veh group (*p* = 0.031) (Figure 1B).

The main treatment [F1,58 = 5.06, *p* = 0.032] and treatment x drug interaction [F1,58 = 9.602, *p* < 0.001] was shown for the average double working memory errors (DWMEs). A significantly increased DWME was observed in the two model groups, Aβ-sham-veh and Aβ-pin-veh (*p* < 0.001). However, melatonin supplementation corrected the impaired memory response in both the Aβ-sham-mel group (*p* < 0.001) and the Aβ-pin-mel group (*p* = 0.016) compared to the C-sham-veh group (Figure 1C).

Finally, the main drug [F1,58 = 19.162, *p* < 0.001] and treatment x drug interaction [F1,58 = 6.062, *p* = 0.001] was found for the time needed to perform the task. While all treatments prolonged the time to reach the criterion (*p* = 0.05, C-pin-veh vs. C-sham-veh; *p* = 0.007, Aβ-sham-veh vs. C-sham-veh; *p* = 0.01, Aβ-pin-veh vs. C-sham-veh), melatonin supplementation significantly shortened the time necessary to complete the task (*p* < 0.05, C-pin-mel vs. C-pin-veh; Aβ-sham-mel vs. Aβ-sham-veh) (Figure 1D).

### 2.2. The Combined Treatment of icvAβ_1-42_ and Pinealectomy Increased Aβ_1-42_ Levels in the Frontal Cortex and Hippocampus

To ensure the efficacy of our model we analyzed the levels of Aβ_1-42_ accumulated in the frontal cortex and hippocampus in all groups of animals tested. Three-way ANOVA demonstrated a main effect of the Aβ-pin treatment for the Aβ_1-42_ levels in the frontal cortex and hippocampus of the rats. The increase in Aβ_1-42_ levels in the frontal cortex was detected only in Aβ-pin-veh rats (*p* < 0.001, compared to C-sham-veh) (Figure 2A). Chronic treatment with melatonin prevented the increase in Aβ_1-42_ in Aβ-pin-veh (*p* < 0.001, Aβ-pin-veh compared to Aβ-pin-mel) (Figure 2A). A similar trend was observed also in the hippocampus, where only the double treatment caused a significant increase in Aβ_1-42_ levels. This was reversed upon melatonin treatment (Figure 2B).

### 2.3. Melatonin Alleviated icvAβ_1-42_ + Pinealectomy-Induced an Increase in GS Levels in the Frontal Cortex

The role of GS in the amyloidogenic pathway of APP processing is essential as it is responsible for the generation of the pathogenic Aβ_1-42_ peptide. Therefore, an increase in GS protein levels leads to an exacerbation of the Aβ_1-42_ burden. To test this, GS levels were measured in the frontal cortex and hippocampus of all animal groups. A frontal cortex elevation of GS was only observed in Aβ-pin-treated rats (*p* < 0.05, C-sham-veh compared to Aβ-pin-veh group) (Figure 3A). Notably, chronic melatonin treatment reduced GS levels to that of controls (*p* < 0.05, Aβ-pin-veh compared to Aβ-pin-mel) (Figure 3A).

In the hippocampus, the double treatment did not cause a significant increase in GS levels. Nevertheless, all three treated animal groups exhibited increased levels of GS compared to the control group (*p* < 0.05, C-sham-veh compared to C-pin-veh, to Aβ-sham-veh and to Aβ-pin-veh group) (Figure 3B). The treatment with melatonin significantly reduced GS levels only in the C-pin treated group (*p* < 0.001, C-pin-veh compared to C-pin-mel group) (Figure 3B).

## 3. Discussion

The progression and potential treatment of AD have been the subject of intense research over the past decade. Several animal models have been used to induce AD, including drug administration, genetic mutations, and Aβ injections. To date, none of these models has fully captured the pathology of AD. For instance, genetic animal models represent only the familial AD, which accounts for a relatively small fraction of AD cases. Moreover, there is a great diversity in the studied mutations in the APP, Tau and/or GS subunits [27], further limiting the applicability of the obtained results, rather than accounting for the vast majority of AD patients who suffer from sporadic AD [28,29]. Hence, another approach is required to induce AD pathology which is common in the majority of cases. Since Aβ accumulation is one of the first hallmarks of AD pathogenesis, intracerebroventricular administration of preformed Aβ_1-42_ aggregated peptides proves adequate in the study of acute Aβ-induced neuroinflammation [30,31]. As summarized in the review by Budni and de Oliveira, 2020 [30], many of these animal models share cognitive and memory decline, as well as increased markers of oxidative stress, neuroinflammation, and apoptosis. These observations are in agreement with our previous work, where we showed that a single icvAβ_1-42_ administration increased lipid peroxidation in the hippocampus and frontal cortex of 10-week-old adult male Sprague-Dawley rats, measured after 40 days [26]. It also caused severe anxiety and memory impairment.

To mimic the sleep disturbance and high oxidative stress levels in the AD brain, we combined icvAβ_1-42_ peptide administration with pinealectomy, i.e., melatonin deficiency. The melatonin deficiency caused by pinealectomy was verified in our previous study [32]. The double model (icvAβ_1-42_ + pin) disturbed the antioxidant systems of superoxide dismutase and glutathione, and aggravated behavioral decline [26].

To compensate for the melatonin deficiency, the animals were treated with exogenous melatonin during the dark phase for 40 days, which significantly lessened the anxiety [26] and restored the memory in both icvAβ_1-42_ and icvAβ_1-42_ + pin groups. Although melatonin supplementation had little effect on superoxide dismutase and glutathione levels, it modified the observed lipid peroxidation as a direct scavenger of free radicals and reactive oxygen species [26].

In the current study, we aimed to substantiate our previous work by examining the effect of icvAβ_1-42_ administration, with and without pinealectomy, on spatial memory, Aβ_1-42_ accumulation, and GS levels in the frontal cortex and hippocampus.

A RAM test was used to measure the animals’ ability to retrieve temporary (working) and stored (reference) memory. Both transgenic and icvAβ_1-42_-induced animal models of AD suffer from a higher degree of WMEs and impaired spatial memory [33,34,35,36]. This memory impediment was corroborated in our animal model. Specifically, pinealectomy alone was sufficient to induce more errors in the RAM test. Similar results were observed for the icvAβ_1-42_ and icvAβ_1-42_ + pin groups. Importantly, melatonin supplementation restored the working memory of the animals to almost the same level as that of the controls.

Regarding Aβ_1-42_ accumulation in the frontal cortex and hippocampus, only the double model showed a significant increase in Aβ_1-42_ levels, which was reversed by melatonin supplementation. This finding supports our hypothesis that melatonin deficiency is essential for Aβ_1-42_ aggregation and accumulation. Consistent with our previous study, we confirm that icvAβ_1-42_ infusion induced an accumulation of the toxic oligomer in both the frontal cortex and hippocampus by increasing the γ-secretase in the hippocampus without affecting α- and β-secretases [37].

The results of the present study are consistent with those of Ali and Kim, 2015 [38], which showed that melatonin reduced the Aβ_1-42_ load and ameliorated icvAβ_1-42_-induced memory impairment. Notably, melatonin had a similar effect on β-amyloid protein levels and cognition in a streptozotocin-induced in vivo AD rat model [39]. In addition, we recently reported that the melatonin analogue, agomelatine, corrects anxiety and memory decline in the RAM test by reducing toxic Aβ_1-42_ and the γ-secretase in the hippocampus [37]. A potential mechanism by which melatonin reduces Aβ_1-42_ levels could be its ability to increase the expression of Transcription Factor EB, which promotes the autophagosome–lysosome clearance of Aβ_1-42_ [40,41]. However, further experiments would be needed to confirm this in our model.

γ-secretase (GS) is an enzyme complex that generates both Aβ_1-40_ and Aβ_1-42_ amyloid peptides. Higher GS activity correlates with Aβ_1-42_ accumulation, and mutations affecting the GS function are found to be responsible for familial AD [1,42]. Thus, inhibition of GS activity has been proposed as a therapeutic strategy against AD progression. Unfortunately, finding promising therapies is challenging because GS is involved in many other cellular processes, such as synaptic plasticity, neuronal excitability, cell adhesion, and intercellular communication [43], and its direct inhibition could have significant side effects [14]. In contrast, GS modulators regulate the enzyme activity rather than fully inhibit it. Although no single modulator has yet been approved, many have been shown to reduce the concentration of Aβ_1-42_ without changing the total amount of Aβ peptides [14].

Here, GS levels in the frontal cortex increased only in the icvAβ_1-42_ + pin group, which is consistent with the high Aβ_1-42_ levels in the same group. In the hippocampus, any manipulation of the animals leads to an increase in GS levels. Notably, melatonin therapy counteracts the increase in GS in both structures and brings GS back to control levels, which is in agreement with the work of Shukla M. et al., 2015 [44], which showed that melatonin inhibited the amyloidogenic processing of APP by stimulating α-secretase and consequently downregulating the gene expression of both β- and γ-secretases. Melatonin has been refuted as a direct modulator of GS [45]. However, the hormone cannot be excluded as a regulator of other cellular pathways involved in GS function. Melatonin provides neuroprotective effects across various brain regions through its MT1 and MT2 receptors, which are abundantly found in areas like the frontal cortex and hippocampus. By activating multiple MT receptor-mediated pathways, melatonin can both reduce Aβ production and enhance its clearance. Several studies have demonstrated that melatonin promotes non-amyloidogenic processing of the APP, leading to reduced Aβ production. In our previous research, we showed that melatonin mitigated the AD-related pathology in a hybrid icvAβ_1-42_ + pin model by stimulating the non-amyloidogenic pathway through both non-receptor mechanisms (SIRT1) and receptor-mediated ERK1/2/CREB signaling in the hippocampus [46]. Based on the findings of the current study, we propose that melatonin’s reduction of Aβ oligomer levels in the frontal cortex may be due to its direct effect on the amyloidogenic pathway by inhibiting γ-secretase activity, whereas in the hippocampus, it likely exerts its protective effects through the non-amyloidogenic pathway, as we have previously reported [46].

Furthermore, studies using male senescence-accelerated OXYS rats, a model for sporadic AD, showed that melatonin alleviated anxiety, maintained reference memory, and decreased Aβ_1-42_ accumulation in the frontal cortex and hippocampus [47]. It is also important to note that AD patients carrying two *APOE4* alleles—a key genetic risk factor—have been found to have nearly half the cerebrospinal fluid melatonin levels compared to those with one allele [48]. Research has shown that melatonin inhibits Aβ aggregation in astrocytes derived from transgenic AD mice overexpressing *APOE4* [49], indicating that exogenous melatonin could be especially beneficial for AD patients with this genetic profile. Though clinical data on melatonin’s effect in AD patients remain somewhat limited, there have been promising results. One clinical trial administered Circadin^®^ (i.e., melatonin) for 24 weeks, resulting in improved cognitive performance and better sleep maintenance [50]. The measured outcomes of these studies are in agreement with our results on melatonin supplementation in the icvAβ_1-42_ + pin model and support melatonin’s therapeutic potential in AD. Based on the aforementioned studies showing the well-known anti-oxidative properties of melatonin and its effect to counteract memory impairments, we can hypothesize that melatonin can exhibit a prophylactic effect and can be administered long before the full manifestation of AD.

During the preparation of this manuscript, we noticed that not many studies using the icvAβ_1-42_ animal model assessed the brain levels of Aβ_1-42_ and GS. Therefore, we believe that our established model is a promising and highly relevant model to study the pathology of AD through the induction of both AD and melatonin deficiency (icvAβ_1-42_ + pin).

## 4. Materials and Methods

### 4.1. Animals

Young adult (10-week-old) male Sprague-Dawley rats (300 g body weight) (Charles River Lab., Wilmington, MA, USA), purchased from the vivarium of the Institute of Neurobiology, BAS, were housed in groups of three to four in clear Plexiglas cages under standard conditions with a 12/12 h light-dark cycle (lights on at 08:00 h) and access to water and laboratory chow ad libitum. Experimental procedures were performed in full compliance with the guidelines of the European Community Council Directive 2010/63/E.U. The animal experiments were approved by the research project (# 300/N◦5888-0183) of the Bulgarian Food Safety Agency.

### 4.2. Experimental Design and Treatment with Melatonin

The experimental design is detailed in Figure 4. Briefly, treatment with melatonin dissolved in 1% hydroxyethylcellulose was started on the same day after intracerebroventricular (icv) infusion of Aβ_1-42_, at a dose of 50 mg/kg, intraperitoneally (i.p.), injected approximately 2 h before the onset of the dark phase for 40 days. The following eight groups used Sham-operated and vehicle-infused and treated rats (C-sham-veh group) (n = 8); pinealectomized and vehicle-infused and treated rats (C-pin-veh) (n = 8); sham-operated, Aβ_1-42_-infused and vehicle-treated rats (Aβ-sham-veh) (n = 8); pin-operated, Aβ_1-42_-infused and vehicle-treated rats (Aβ-pin-veh) (n = 8), sham-operated and vehicle-infused rats and treated with melatonin rats (C-sham-mel group) (n = 8); rats with pinealectomy, infused with the vehicle and treated with melatonin (C-pin-mel group) (n = 8), sham-operated, Aβ_1-42_ infused and treated with melatonin (Aβ-sham-mel group) (n = 8); pin-operated, Aβ_1-42_-infused and treated with melatonin (Aβ-pin-mel group) (n = 8).

### 4.3. Surgery and icv Injection of Aβ_1-42_

The procedure of the surgery and Aβ_1-42_ injection was described in detail in our previous work [26]. Briefly, rats were placed on a stereotaxic apparatus (Stoelting, Wood Dale, IL, USA) under deep anesthesia. After removing the pineal gland with thin forceps, two cannulas were implanted bilaterally according to the atlas of Paxinos and Watson [51] at the following coordinates: (AP = −0.8, L = ±1.5, H = 3.8). Amyloid β_1-42_ (100 µg; FOT Ltd., Sofia, Bulgaria) was prepared as previously described [26] to generate neurotoxic fibrils. The Aβ_1-42_ infusion was performed icv with a 5 mL Hamilton microsyringe at a rate of 1 mL/min for 5 min. Phosphate buffered saline was infused in the sham-operated group. A few days after surgery, the rats were injected with lactated Ringer’s solution and an antibiotic (gentamicin, s.c.).

### 4.4. Radial Arm Maze Test

The spatial hippocampus-dependent learning and memory response was assessed using an 8-arm radial maze (RAM) (Harvard Bioscience Inc., Holliston, MA, USA) between 10:00 a.m. and 12:00 p.m. in a separate soundproof room under artificially diffused light, where the rats were accommodated for 30 min before the test. The rats were also handled repeatedly for 5 days prior to the behavioral procedures to reduce the effect of stress. The RAM test was performed as described in our recent report [52]. The apparatus consisted of eight stainless steel arms (42 × 12 × 12) raised 50 cm above the floor. A number of different photographs (triangle, circle, square) were placed around the apparatus to provide spatial orientation for the rats during the procedure. Seven days before the test, the rats were deprived of food to reach approximately 15% of their body weight, and three days were spent in a shaping procedure to acclimate the rats to the apparatus, in which a random number of food pellets were scattered in each of the eight arms for 15 min. The test lasted 5 days (one session per day). One pellet was placed at the end of each arm. The following parameters were recorded: number of working memory errors (WMEs) per each session over 5 days; average WMEs per session; average number of double WMEs (DWMEs) per session; average time taken to consume all the pellets per session for up to 10 min.

### 4.5. Detection of Biochemical Markers in the Homogenates from the Frontal Cortex and Hippocampus

After behavioral tests, all rats were decapitated with a guillotine after mild anesthesia with CO_2_ and the whole brains were removed. Frontal cortexes and hippocampi were carefully excised. The homogenates were prepared as described previously [26]. To account for the intricate composition of the samples, tissue homogenization was carried out under standardized conditions to ensure uniformity across samples. In addition, all samples were homogenized together to minimize handling differences. Protease inhibitors were added to prevent protein degradation, and all samples were separated into aliquots and stored under conditions that minimized freeze–thaw cycles and degradation. Initially, the samples were tested at a serial dilution, to ensure that the measured levels fell within the range of the assay, avoiding saturation or excessive dilution. Additionally, each plate included a standard curve that was used to calculate the protein concentration. The quality of the standard curve was assessed through the coefficient of determination (R^2^), which was consistently above 0.98. All samples were tested in duplicates.

#### 4.5.1. Determination of Aβ_1-42_ Levels

Quantitative analysis of Aβ_1-42_ levels was performed with Rat Aβ_1-42_ ELISA kit (Cat. No MBS 726579, MyBioSource, Inc., San Diego, CA 92195-3308, USA) according to the manufacturer’s instructions. The absorbance was measured using a microplate reader (Tecan Infinite F200 PRO (Tecan Trading GmbH, Männedorf, Switzerland) at a wavelength of 450 nm.

#### 4.5.2. Determination of GS Levels

Quantitative analysis of γ-secretase levels was performed with a Rat γ-secretase ELISA kit (Cat. No MBS 1600385, MyBioSource, Inc., San Diego, CA 92195-3308, USA) according to the manufacturer’s instructions. The absorbance was measured using a microplate reader (Tecan Infinite F200 PRO (Tecan Trading GmbH, Männedorf, Switzerland) at a wavelength of 450 nm.

### 4.6. Statistical Analysis

Three-way ANOVA was used to analyze the behavioral and biochemical data with factors: pinealectomy (sham and pin), Aβ (control, Aβ), and drug (vehicle, melatonin) followed by a post hoc Bonferroni test in the case of a detected significant difference (SigmaStat 11.0; Palo Alto, CA, USA). When data were not homogenously distributed, nonparametric tests were applied (a Kruskal–Wallis on ranks followed by a Mann–Whitney U test). The significant level was set at *p* ≤ 0.05.

## 5. Conclusions

In conclusion, we have further supported the oxidative stress-induced icvAβ_1-42_ + pin in vivo model to study Alzheimer’s disease, which encompasses many of the neurological and behavioral changes associated with the disease. Therefore, it represents the sporadic AD which affects the majority of AD patients, unlike many widely used transgenic animal models, which mimic only familial AD. In the present work, we show that the combined induction of AD pathology through icvAβ_1-42_ oligomer administration and melatonin deficiency via pinealectomy increases Aβ_1-42_ accumulation and elevates γ-secretase levels in the frontal cortex and hippocampus. This extends our previous findings where we show that the icvAβ_1-42_ + pin animal model causes pronounced anxiety, cognitive decline, and memory impairment, and triggers oxidative stress. Furthermore, we showed that melatonin treatment minimizes the negative effects of the icvAβ_1-42_ infusion and pinealectomy on the amyloidogenic pathway and behavioral state during the development of the in vivo AD model. We are confident that our model offers a valuable tool for understanding AD pathogenesis and for exploring other potential targets and therapies against AD.

## Figures and Tables

**Figure 1 ijms-25-10294-f001:**
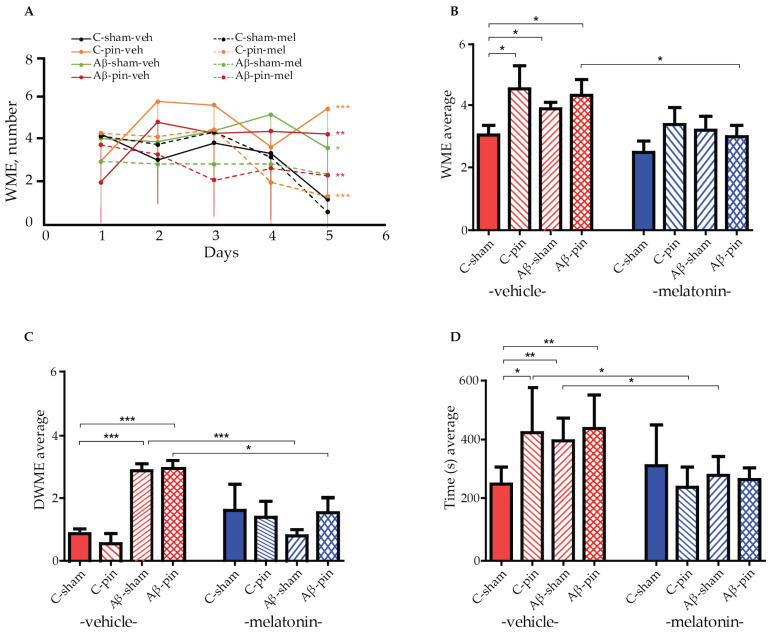
The effect of pinealectomy (pin) and chronic treatment with melatonin (mel) on icvAβ_1-42_ -related effect on (**A**) mean working memory errors (WMEs) during each trial 1st to 5th, (**B**) the average WMEs, (**C**) the average double WMEs (DWMEs), and (**D**) the average time to fulfill the criterion. Data are presented as mean ± S.E.M. (**A**) *** *p* < 0.001, C-pin-veh vs. C-sham-veh group; * *p* = 0.026, Aβ-sham-veh vs. C-sham-veh; ** *p* < 0.01, Aβ-pin-veh vs. C-sham-veh; *** *p* < 0.001, C-pin-mel vs. C-pin-veh; ** *p* = 0.005, Aβ-pin-mel vs. Aβ-pin-veh; (**B**) * *p* = 0.018, C-pin-veh vs. C-sham-veh; * *p* = 0.048, Aβ-sham-veh vs. C-sham; *p* = 0.05, Aβ-pin-veh vs. C-sham; *p* = 0.031, Aβ-pin-mel vs. Aβ-pin-veh; (**C**) *** *p* < 0.001, Aβ-sham-veh and Aβ-pin-veh vs. C-sham-veh group; *** *p* < 0.001, Aβ-sham-mel vs. Aβ-sham-veh; * *p* = 0.016, Aβ-pin-mel vs. Aβ-pin-veh; (**D**) * *p* = 0.05, C-pin-veh vs. C-sham-veh; ** *p* = 0.007, Aβ-sham-veh vs. C-sham-veh; ** *p* = 0.01, Aβ-pin-veh vs. C-sham-veh; * *p* < 0.05, C-pin-mel vs. C-pin-veh; ** *p* = 0.01, Aβ-sham-mel vs. Aβ-sham-veh.

**Figure 2 ijms-25-10294-f002:**
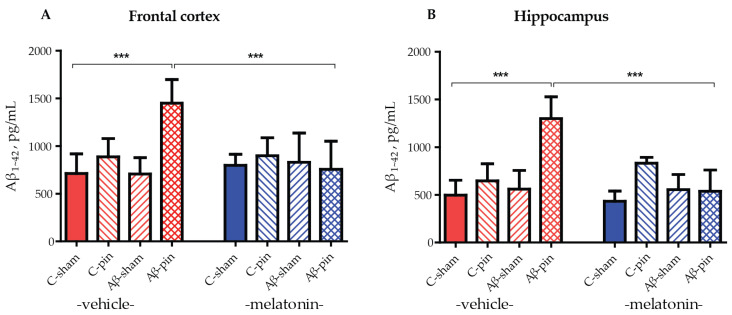
Aβ_1-42_ levels (pg/mL) under the influence of icvAβ_1-42_ (Aβ) and/or pinealectomy (pin) and chronic treatment with melatonin (mel) in the (**A**) frontal cortex and (**B**) hippocampus. Data are presented as mean ± S.E.M. (**A**) *** *p* < 0.001, C-sham-veh vs. Aβ-pin-veh group and Aβ-pin-veh vs. Aβ-pin-mel group. (**B**) *** *p* < 0.001, C-sham-veh vs. Aβ-pin-veh group; Aβ-pin-veh vs. Aβ-pin-mel group.

**Figure 3 ijms-25-10294-f003:**
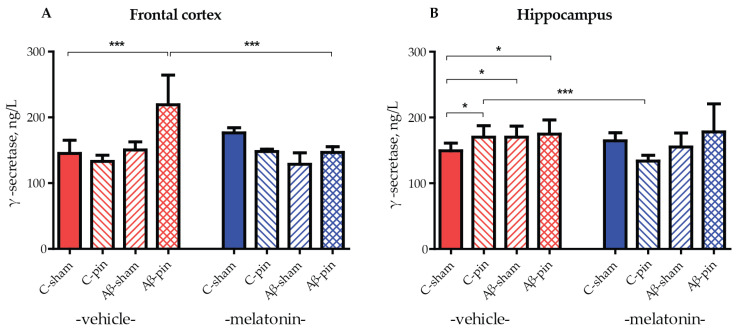
The influence of icvAβ_1-42_ (Aβ) and/or pinealectomy (pin) and chronic treatment with melatonin (mel) on γ-secretase (GS) levels (ng/L) in the (**A**) frontal cortex and (**B**) hippocampus. Data are presented as mean ± S.E.M. (**A**) *** *p* < 0.001, C-sham-veh compared to Aβ-pin-veh group and Aβ-pin-veh compared to Aβ-pin-mel group. (**B**) * *p* < 0.05, C-sham-veh compared to C-pin-veh; C-sham-veh compared to Aβ-sham-veh; C-sham-veh compared to Aβ-pin-veh group; *** *p* < 0.001, C-pin-veh compared to C-pin-mel group.

**Figure 4 ijms-25-10294-f004:**
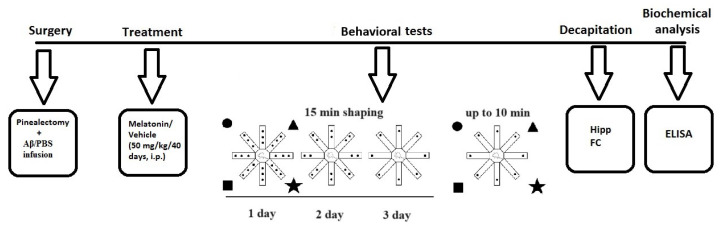
Schematic diagram of the experimental protocol. Brain structures used to determine biochemical markers were taken from the cohort of rats studied in the radial arm maze test.

## Data Availability

The data that support the findings of this study are available from the corresponding author, R.T., upon reasonable request.

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
