# Peer review of "Melatonin Supplementation Alleviates Impaired Spatial Memory by Influencing Aβ1-42 Metabolism via γ-Secretase in the icvAβ1-42 Rat Model with Pinealectomy"

_ijms, 2024, doi:10.3390/ijms251910294_

Round 1

Reviewer 1 Report

Comments and Suggestions for Authors

Dear Authors,

recommendations for the revision:

Specific Comments:

·       In the abstract: Authors have mentioned that “To better model sporadic AD, we introduced a new AD rat model induced by intracerebroventricular administration of Aβ1-42 oligomers (icvAβ1-42) combined with melatonin deficiency via pinealectomy (pin).” However, authors have already used this model in their previous manuscript entitled “The Role of Melatonin on Behavioral Changes and Concomitant Oxidative Stress in icvAβ1-42 Rat Model with Pinealectomy”. Is the model used in the current manuscript different than the previous article?

·       How have the authors verified the complete removal of the pineal gland? Did you measure any biochemical parameters or perform histological sectioning to confirm total Pinealectomy?

·       What could be the possible mechanism for the regulation of γ-secretase (GS) by melatonin in the frontal cortex and Hippocampus, as melatonin reduced GS ONLY in the Aβ-pin group in the frontal cortex whereas ONLY in the C-pin group in Hippocampus?

·       In the method section: Mention the Sex of animals used in the study: Male or female??

Author Response

Answers to Reviewer 1:

Reviewer 1 wrote:

  • In the abstract: Authors have mentioned that “To better model sporadic AD, we introduced a new AD rat model induced by intracerebroventricular administration of Aβ1-42 oligomers (icvAβ1-42) combined with melatonin deficiency via pinealectomy (pin).” However, authors have already used this model in their previous manuscript entitled “The Role of Melatonin on Behavioral Changes and Concomitant Oxidative Stress in icvAβ1-42 Rat Model with Pinealectomy”. Is the model used in the current manuscript different than the previous article?
  • How have the authors verified the complete removal of the pineal gland? Did you measure any biochemical parameters or perform histological sectioning to confirm total Pinealectomy?
  • What could be the possible mechanism for the regulation of γ-secretase (GS) by melatonin in the frontal cortex and Hippocampus, as melatonin reduced GS ONLY in the Aβ-pin group in the frontal cortex whereas ONLY in the C-pin group in Hippocampus?
  • In the method section: Mention the Sex of animals used in the study: Male or female??

Answer to Reviewer 1:

We appreciate the Reviewer’s comments and suggestions and we have addressed them.

Point #1

In the abstract: Authors have mentioned that “To better model sporadic AD, we introduced a new AD rat model induced by intracerebroventricular administration of Aβ1-42 oligomers (icvAβ1-42) combined with melatonin deficiency via pinealectomy (pin).” However, authors have already used this model in their previous manuscript entitled “The Role of Melatonin on Behavioral Changes and Concomitant Oxidative Stress in icvAβ1-42 Rat Model with Pinealectomy”. Is the model used in the current manuscript different than the previous article?

Response: The same model was used in the current paper. The aim of the current paper is to further support our findings by determining the actual levels of Aβ1-42 and γ-secretase. We find this necessary as not many papers evaluate the effect of icvAβ1-42 administration on Aβ1-42 accumulation. As we do not wish to mislead the readers, we stated in the final paragraph of the introduction our previous research on the model (Lines 87-92: “Previously, we introduced a new rat model of AD with induced melatonin deficiency through pinealectomy (pin) and intracerebroventricular (icv) infusion of Aβ1-42 oligomer (Tzoneva et al., IJMS 2021). We demonstrated that the combination of pin and icvAβ1-42 provoked significant changes in anxiety and oxidative stress in the animals compared to either pin or icvAβ1-42 groups alone. Moreover, exogenous melatonin treatment of the rat AD model attenuated the observed behavioral changes and oxidative stress elevation.”) and how we plan to expand upon it (Lines 92-96: In the present study, we aimed to validate further our established animal model of AD (icvAβ1-42 + pin) and confirm that it worsens spatial memory in rats and enhances Aβ1-42 production via increased γ-secretase levels in the frontal cortex and hippocampus. We also tested whether melatonin supplementation could correct these alterations.) We further explained this in the Methods section Lines 338-339 “The procedure of surgery and Aβ1-42 injection was described in detail in our previous work (Tzoneva et al., IJMS 2021)…”. We also redacted the abstract to Lines 16-17: “To better model sporadic AD, we USED a new AD rat model induced by intracerebroventricular…” and the conclusion – Lines 408-409: “This extends our PREVIOUS findings where we show that icvAβ1-42 + pin animal model causes pronounced…” As we have yet to see the implementation of the icvAβ1-42 + pin model by other researchers we consider it still new in the field.

Point #2

How have the authors verified the complete removal of the pineal gland? Did you measure any biochemical parameters or perform histological sectioning to confirm total Pinealectomy?

Response: Surgical removal of the pineal gland is routinely used in our lab and studies using this model have been published in several journals (Tchekalarova et al., 2016, 2018, 2019, 2020, 2021, 2023, 2024). We have recently reported low plasma hormone levels and changes in MT (melatonin) receptors as a consequence of melatonin deficiency. In addition, pinealectomy is associated with progressive neuronal loss found in specific brain structures (Tchekalarova J., et al., Behav Brain Res. 2016 doi: 10.1016/j.bbr.2015.12.043). Thus, plasma melatonin levels in sham-operated rats have a circadian pattern with a peak during the dark period and a nadir during the light period (Tchekalarova J. et al., Neuroscience Letters 2020 doi: 10.1016/j.neulet.2019.1346371). The pineal gland is the primary source of the hormone, while several other tissues, particularly the gastrointestinal tract, release the paracrine hormone (H.F. Urbanski, Neuroendocrinol. Physiol. Med. 2000 doi: 10.1007/978-1-59259-707-9_23). However, the melatonin secreted by the gastrointestinal tract does not reach the brain and has only local functions such as such as regulating intestinal motility and influencing gut immune function (Chen CQ, et al., World J Gastroenterol. 2011 doi: 10.3748/wjg.v17.i34.3888). In addition, we reported that rats with pinealectomy had a flattened 24-hour pattern and low plasma melatonin levels. In this respect we added in Lines 220-221 the following sentence: The melatonin deficiency caused by pinealectomy was verified in our previous study [Tchekalarova J, et al., Behav Brain Res. 2016 doi: 10.1016/j.bbr.2015.12.043].

Point #3

What could be the possible mechanism for the regulation of γ-secretase (GS) by melatonin in the frontal cortex and Hippocampus, as melatonin reduced GS ONLY in the Aβ-pin group in the frontal cortex whereas ONLY in the C-pin group in Hippocampus?

Response: Thank you for the question. Melatonin exerts neuroprotective effects throughout the brain via MT1 and MT2 receptors, which are widely distributed, including in the cerebral cortex and hippocampus. The hormone is capable of triggering multiple MT receptor-mediated signaling pathways leading to decreased Aβ production and increased Aβ clearance. Several studies have shown that melatonin could reduce Aβ production by enhancing non-amyloidogenic APP processing. Recently, we also reported that melatonin attenuated AD-related pathogenesis in the same hybrid Aβ + pin model by stimulating the non-amyloidogenic pathway via both non-receptor (SIRT1) and MT1A and receptor-related ERK1/2/CREB signaling in the hippocampus (Tchekalarova et al., 2024, doi: 10.3390/ijms25031867). From the results of the present study, we could speculate that melatonin's attenuating effect on Aβ oligomer levels in the frontal cortex is a direct protective effect on the amyloidogenic processing pathway via suppression of γ-secretase activity, whereas in the hippocampus the hormone acts through a non-amyloidogenic pathway, as we have previously reported (Tchekalarova J., et al., IJMS 2024 doi: 10.3390/ijms25031867). We have commented on this in the discussion Lines 276-289.

Point #4

In the method section: Mention the Sex of animals used in the study: Male or female??

Response: Thank you for this note. We inserted in the text the sex of animals used (Line 313). We used male rats only.

We thank the Reviewer for carefully assessing the manuscript and the valuable recommendations and clarifications. We truly believe that it immensely improved the clarity and quality of the review.

Reviewer 2 Report

Comments and Suggestions for Authors

The authors in this paper describe a new model of AD in the rat, obtained following intracerebroventricular administration of Aβ1-42 oligomers combined with melatonin deficiency via pinealectomy. Certainly, the topic discussed is of great interest since it could help to clarify certain aspects of sporadic AD and deserves to be published.

The topic is well presented, clear and straightforward, supported by the literature. The literature references are adequate and recent.

In this paper as described in the discussion section, spatial memory is assessed with the radial arm maze test and Aβ1-42 and γ-secretase levels are measured in the frontal cortex and hippocampus with the ELISA method. The data are presented in a clear and statically appropriate manner.

The results presented show that the icvAβ1-42 + pin model had a reduction in spatial memory and increased levels of Aβ1-42 and γ-secretase in the frontal cortex and hippocampus, effects not observed with either icvAβ1-42 or pin alone.

Very interesting are the studies carried out after melatonin administration, this treatment reversed memory deficits and reduced Aβ1-42 and γ-secretase levels in both structures.

Some curiosities for the authors:

1) Have you considered performing electron microcopy studies to visualize in tissue the decrease in Aβ1-42 oligomers after melatonin administration?

2) Have you thought of carrying out immunohistochemistry studies to assess in tissue the decrease in Aβ1-42 levels after melatonin administration?

Author Response

Answers to Reviewer 2:

Reviewer 2 wrote:

The authors in this paper describe a new model of AD in the rat, obtained following intracerebroventricular administration of Aβ1-42 oligomers combined with melatonin deficiency via pinealectomy. Certainly, the topic discussed is of great interest since it could help to clarify certain aspects of sporadic AD and deserves to be published.

The topic is well presented, clear and straightforward, supported by the literature. The literature references are adequate and recent.

In this paper as described in the discussion section, spatial memory is assessed with the radial arm maze test and Aβ1-42 and γ-secretase levels are measured in the frontal cortex and hippocampus with the ELISA method. The data are presented in a clear and statically appropriate manner.

The results presented show that the icvAβ1-42 + pin model had a reduction in spatial memory and increased levels of Aβ1-42 and γ-secretase in the frontal cortex and hippocampus, effects not observed with either icvAβ1-42 or pin alone.

Very interesting are the studies carried out after melatonin administration, this treatment reversed memory deficits and reduced Aβ1-42 and γ-secretase levels in both structures.

Some curiosities for the authors:

1) Have you considered performing electron microcopy studies to visualize in tissue the decrease in Aβ1-42 oligomers after melatonin administration?

2) Have you thought of carrying out immunohistochemistry studies to assess in tissue the decrease in Aβ1-42 levels after melatonin administration?

Answer to Reviewer 2:

We are grateful to the Reviewer for his kind and positive opinion of our manuscript. We highly appreciate the given suggestions. Indeed, we find the curious.

Point #1

Have you considered performing electron microcopy studies to visualize in tissue the decrease in Aβ1-42 oligomers after melatonin administration?

Response: Thank you for the suggestion. We are considering further experiments in this regard. One reason is that according to literature data Aβ1-42 monomers, oligomers and fibers could have different effects on neurovascular units (Georgieva I. et al. IJMS 2023 doi: 10.3390/IJMS241411344). Another is that it would be of great value to know which type of Aβ1-42 structures are reduced the most upon melatonin treatment since an ELISA assay cannot provide us with such information.

Point #2

Have you thought of carrying out immunohistochemistry studies to assess in tissue the decrease in Aβ1-42 levels after melatonin administration?

Response: Thank you for the suggestion. So far we have used immunohistochemistry to validate the neuronal loss in rats with pinealectomy (Tchekalarova et al., Behavioural Brain Research 2016 doi: 10.1016/j.bbr.2015.12.043) and separately in icvAβ1-42-induced rat model of AD (Ilieva et al., Physiol. Behav. 2021 doi: 10.1016/j.physbeh.2021.113525). In the near future we have considered immunohistochemical analysis to investigate the distribution of Aβ1-42 in the brain and to assess more directly both the effect of icvAβ1-42 and of melatonin treatment. This will certainly promote our icvAβ1-42 + pin model and improve our understanding of melatonin’s function in AD.

We would like to thank the Reviewer for his comments and we believe that our future work will be improved due to them.

Reviewer 3 Report

Comments and Suggestions for Authors

This study examines the effects of melatonin supplementation on spatial memory impairment in a rat model of Alzheimer's disease (AD). The impairment is induced by administering amyloid-beta (Aβ1-42) peptides directly into the brain's ventricles and removing the pineal gland to simulate a deficiency in melatonin. The study emphasizes the involvement of γ-secretase in the metabolism of Aβ1-42 and investigates if melatonin can alleviate the memory impairments and metabolic alterations linked to the course of Alzheimer's disease.

Here are some critical scientific questions for the manuscript:

1.        How can the effects of pinealectomy alone be distinguished from its conjunction with Aβ1-42 injection in terms of spatial memory impairment and Aβ1-42 metabolism?

2.        Can you provide details on the age and sex of the rats used in the study, considering these factors can significantly influence the outcome of AD models?

3.        How was the dose of melatonin chosen for this study, and is there evidence from prior research that supports its effectiveness in similar AD models?

4.        Considering the multifaceted nature of γ-secretase, a sophisticated enzyme implicated in various physiological mechanisms, how can you mitigate the possible unintended impacts of melatonin on unrelated pathways, which could potentially affect Aβ1-42 metabolism?

5.        According to the study, melatonin administration is said to counteract memory impairments. Could you perhaps specify whether melatonin exhibits any prophylactic effects or if it solely mitigates the impairments once they have manifested?

6.        What are the precise functions of the various constituents of γ-secretase in the observed disease development in your model, and how does melatonin impact each constituent?

7.        Was there an assessment of inflammatory markers or oxidative stress levels in addition to the analysis of Aβ1-42 and γ-secretase levels, in order to enhance the knowledge of the model's pathology?

8.        Are there any enduring consequences of melatonin administration detected beyond the acute recovery of memory impairments, specifically in relation to the development of amyloid plaques and neurodegeneration in this model?

9.        How do the results of this study compare to previous research on the impact of melatonin in human patients with Alzheimer's disease or other animal models of Alzheimer's disease?

10.   The manuscript discusses the utilization of ELISA to quantify Aβ1-42 and γ-secretase concentrations. How can you guarantee the accuracy and precision of these measures, particularly considering the intricate composition of the samples?

11.   Considering the intricate nature of the development of Alzheimer's disease (AD), which involves various routes and variables, how accurately does the icvAβ1-42 + pin model reflect the sporadic type of AD that is commonly observed in humans?

12.   Would the study be enhanced by include other behavioral tests to evaluate cognitive functions impacted by AD, such as anxiety or depressive-like behaviors, which are frequently observed in AD patients?

13.   The introduction asserts that sporadic AD generally originates from stress factors such as oxidative stress. However, the experimental emphasis appears to be predominantly on the metabolism of amyloid-beta. How do you harmonize these variables while formulating your opinions and suggestions for AD therapy?

Author Response

Answer to Reviewer 3:

We thank the Reviewer for the comments and suggestions and have made efforts to address them fully. We shall address them one by one.

Point #1

How can the effects of pinealectomy alone be distinguished from its conjunction with Aβ1-42 injection in terms of spatial memory impairment and Aβ1-42 metabolism?

Response: Pinealectomy per se, as well as β-amyloid, impaired spatial memory in the RAM test. These data are consistent with previous reports (Özkaya et al., 2014; Ilieva et al., 2019). In the present study, we reported that the memory-enhancing effect of either pinealectomy or β-amyloid was not enhanced in the hybrid model, although melatonin supplementation successfully corrected the memory impairment. However, the hybrid model produced an effect on β-amyloid expression and γ-secretase in the FC, unlike pinealectomy and β-amyloid, suggesting that the hybrid model is more reliable for inducing the pathogenesis associated with AD.

Point #2

Can you provide details on the age and sex of the rats used in the study, considering these factors can significantly influence the outcome of AD models?

Response: We are thankful for this note. Indeed, age factor affected memory responses in rats with pinealectomy. And these results are given in the preprint manuscript (Tchekalarova J., et al., Preprints.org 2024 doi: 10.20944/preprints202407.1486.v1). We used young adult (ten-week-old) male rats only and did not study sex-dependent effects. We have added this information in section 4.1. Animals Line 313. For example, the 5xFAD transgenic mice model of AD starts to show symptoms of Aβ1-42 accumulation, cognitive deficits and neuronal loss at a similar age (Pádua, M.S. et al., Int. J. Mol. Sci. 2024 doi: 10.3390/ijms25126766). Other studies utilizing a similar approach to model sporadic AD in rats have also used young adult animals (Petrasek, T. et al. Front. Aging Neurosci. 2016 doi:10.3389/FNAGI.2016.00083/BIBTEX.; Zhang, J.; et al. PLoS One 2013 doi:10.1371/JOURNAL.PONE.0075786.)

Point #3

How was the dose of melatonin chosen for this study, and is there evidence from prior research that supports its effectiveness in similar AD models?

Response: The dose of melatonin was chosen based on prеvious literature (Labban et al., Degener. Neurol. Neuromuscul. Dis. 2021 doi: 10.2147/DNND.S291172; Yang et al., Sleep Med. Rev. 2020 doi: 10.1016/j.smrv.2019.101235) as well as our study (Tzoneva R, et al. Int J Mol Sci. 2021 doi: 10.3390/ijms222312763) and have proven to have an effect in neurodegenerative diseases and their models.

Point #4

Considering the multifaceted nature of γ-secretase, a sophisticated enzyme implicated in various physiological mechanisms, how can you mitigate the possible unintended impacts of melatonin on unrelated pathways, which could potentially affect Aβ1-42 metabolism?

Response: As we have stated there is no evidence that melatonin acts as a γ-secretase inhibitor, thus we hypothesize that it could modulate its activity. Considering that melatonin supplementation improves memory and cognition in the tested animal groups, we believe that any other signaling pathways, affected by it, do not cause harm to the animals. Since we measure the accumulation of Aβ1-42 and observe a change only in the icvAβ1-42 + pin group currently we can only assume that melatonin acts positively on lowering Aβ1-42 levels. One possible mechanism could be by increased expression of Transcription factor EB by melatonin, which promotes autophagosome-lysosome clearance of Aβ1-42 (Li M, et al. J Pineal Res. 2016 doi: 10.1111/jpi.12353; Xiao Q, et al. J Neurosci. 2014 doi: 10.1523/JNEUROSCI.3788-13.2014). To be certain whether this is the case for our model would require further experiments. We have commented on this potential mechanism in Lines 254-257.

Point #5

According to the study, melatonin administration is said to counteract memory impairments. Could you perhaps specify whether melatonin exhibits any prophylactic effects or if it solely mitigates the impairments once they have manifested?

Response: We did not perform studies with melatonin pretreatment in this hybrid model. Therefore, from the results of the study we can conclude that melatonin only alleviated model-related behavioral impairments. However, due to the link between anti-oxidative properties of melatonin and its established effect to counteract memory impairments we can admit that it could have and positive prophylactic effect in prevention the development of AD. Therefore, we have added an additional text to Lines 303-306: “Based on the aforementioned studies showing the well-known anti-oxidative properties of melatonin and its effect to counteract memory impairments, we can hypothesize that melatonin can exhibit a prophylactic effect and can be administered far before the full manifestation of AD.”

Point #6

What are the precise functions of the various constituents of γ-secretase in the observed disease development in your model, and how does melatonin impact each constituent?

Response: Thank you for the valuable note. For this study we used a commercial ELISA kit for γ-secretase detection, which most likely targets the matured form of the enzyme where presenilin (1 or 2) : nicastrin : APH-1 : and PEN-2 are in 1:1:1:1 ratio. Typically, such ELISA kits target the APH-1 subunit as it forms the “core” of the active enzyme. Unfortunately, we do not have exact information of the specific antigen target used in the applied ELISA kit and our attempts to receive this information were fruitless. Therefore, we can only comment on the total amount of γ-secretase and measure the levels of each subunit in the future by immunoprecipitation and/or western blot.

Point #7

Was there an assessment of inflammatory markers or oxidative stress levels in addition to the analysis of Aβ1-42 and γ-secretase levels, in order to enhance the knowledge of the model's pathology?

Response: In our previous paper, where we first introduced the icvAβ1-42 + pin model, we measured the levels of superoxide dismutase (SOD), total glutathione (GSH) and malondialdehyde (MDA) in the same animal groups. We discovered that the icvAβ1-42 + pin model suffered from significant changes in the antioxidant systems of GSH and SOD, which resulted in increased MDA levels compared to the controls. This correlated with increased anxiety determined by the open field and elevated maze tests. Additionally, the model suffered from reduced recognition memory. We have referred to this in Lines 88-93 „Previously, we introduced a new rat model of AD with induced melatonin deficiency through pinealectomy (pin) and intracerebroventricular (icv) infusion of Aβ1-42 oligomer (Tzoneva et al., 2021). We demonstrated that the combination of pin and icvAβ1-42 provoked significant changes in anxiety and oxidative stress in the animals compared to either pin or icvAβ1-42 groups alone. Moreover, exogenous melatonin treatment of the rat AD model attenuated the observed behavioral changes and oxidative stress elevation” and Lines 225-230 “To compensate for the melatonin deficiency, the animals were treated with exogenous melatonin during the dark phase for 40 days, which significantly lessened the anxiety [26] and restored the memory in both icvAβ1-42 and icvAβ1-42 + pin groups. Although melatonin supplementation had little effect on superoxide dismutase and glutathione levels, it modified the observed lipid peroxidation as direct scavenger of free radicals and reactive oxygen species [26].” In the present manuscript we show that elevated levels of Aβ1-42 and γ-secretase correlate with the oxidative stress measured in the icvAβ1-42 + pin model. Additionally, these metabolic changes resonate with decreased spatial, working and reference memory which are common struggles for AD patients.

Point #8

Are there any enduring consequences of melatonin administration detected beyond the acute recovery of memory impairments, specifically in relation to the development of amyloid plaques and neurodegeneration in this model?

Response: Thank you for your comment. In our model, the animals were tested 40 days after icvAβ1-42 administration and/or pinealectomy. Since the icvAβ1-42 + pinealectomy model has not been used as an AD model by other research groups, there is no existing literature that would allow us to form a clear hypothesis. However, based on our previous studies (Tchekalarova et al., Behav. Brain Res. 2016; Tzoneva et al., IJMS 2021) and others (Tasdemir et al., Eur. Rev. Med. Pharmacol. Sci. 2012), we know that the timing of testing after pinealectomy affects the levels of antioxidant markers. This suggests a difference in the biological response to short-term versus long-term melatonin deficiency. Nevertheless, this knowledge alone is insufficient to predict the long-term outcomes of the intervention, including the effects of chronic melatonin supplementation in our model. Given the increasing use of melatonin supplementation in today's society and the potential for studies investigating over-the-counter drugs or natural hormones to be misinterpreted as treatments that do not require medical oversight, we recognize the importance of further investigating this issue. We plan to address this in future research.

Point #9

How do the results of this study compare to previous research on the impact of melatonin in human patients with Alzheimer's disease or other animal models of Alzheimer's disease?

Response: Thank you for the question. In general, our findings are consistent with numerous studies using both transgenic and sporadic models of Alzheimer's disease. Overall, exogenous melatonin supplementation has been shown to improve memory deficits, enhance learning ability, and reduce Aβ1-42 levels by both inhibiting its production and promoting its clearance, as highlighted in a recent meta-analysis (Zhai Z, et al. Neuroscience. 2022 doi: 10.1016/j.neuroscience.2022.09.012) (Added to Introduction, Lines 79-82). In addition, research using a sporadic AD model with male senescence-accelerated OXYS rats demonstrated that melatonin mitigated anxiety, preserved reference memory, and reduced Aβ1-42 accumulation in the frontal cortex and hippocampus (Rudnitskaya EA et al. J Alzheimers Dis. 2015 doi: 10.3233/JAD-150161). Clinical data on the effects of melatonin in AD patients, while limited, are promising. For instance, a clinical trial (NCT00940589, 2009–2013) administered 2 mg of Circadin® (melatonin) to AD patients over 24 weeks, showing improvements in cognitive function and sleep maintenance (Wade AG, et al. Clin Interv Aging. 2014 doi: 10.2147/CIA.S65625). Additionally, another trial (NCT04522960, 2020–2023) aimed to compare melatonin levels in AD patients and healthy individuals, correlating these levels with cognitive function and blood/CSF concentrations, although results have yet to be released. Currently, two ongoing clinical trials are further investigating melatonin’s impact on AD: NCT03954899 (2019–2025), which is assessing whether 9 months of melatonin treatment can improve AD biomarkers (e.g., hyperphosphorylated tau and Aβ1-42) and cognitive function in patients with mild cognitive impairment, and NCT05629871 (2023–2027), which aims to confirm the relationship between Aβ peptide accumulation and sleep disturbances. These studies will hopefully provide valuable insights into melatonin's therapeutic potential. Moreover, a meta-analysis of AD patients who received melatonin for more than 12 weeks demonstrated improvements in Mini-Mental State Examination scores, further supporting its cognitive benefits (Sumsuzzman DM, et al. Neurosci Biobehav Rev. 2021 doi: 10.1016/j.neubiorev.2021.04.034). It is also worth mentioning that while AD patients carrying two APOE4 alleles (risk factor of sporadic AD) have been found to have approximately half the cerebrospinal fluid melatonin levels compared to those with one allele (Liu RY, et al. J Clin Endocrinol Metab. 1999 doi: 10.1210/jcem.84.1.5394), melatonin has been shown to inhibit Aβ aggregation in astrocytes cultured from transgenic AD mice overexpressing Apoe4 (Poeggeler B, et al. Biochemistry. 2001 doi: 10.1021/bi0114269). This suggests that exogenous melatonin may be particularly beneficial for AD patients with this genetic risk factor. We have added the answer to the manuscript Lines 291-307. We have also rearranged the discussion and introduction sections in order to provide a better introduction to the issue of melatonin deficiency in AD patients (Lines 211-218 from the Discussion are moved to Lines 74-84 in the Introduction.)

Point #10

The manuscript discusses the utilization of ELISA to quantify Aβ1-42 and γ-secretase concentrations. How can you guarantee the accuracy and precision of these measures, particularly considering the intricate composition of the samples?

Response: Thank you for this question. Indeed, the tissue samples that we used for this and prior studies are complex. To account for this tissue homogenization was carried out under standardized conditions to ensure uniformity across samples. Also all samples were homogenized together to minimize handling differences. Protease inhibitors were added to prevent protein degradation, and all samples were separated into aliquots and stored under conditions that minimized freeze-thaw cycles and degradation. Initially the samples were tested at a serial dilution, to ensure that the measured levels fall within the range of the assay, avoiding saturation or excessive dilution. Additionally, each plate included a standard curve that was used to calculate the protein concentration. The quality of the standard curve was assessed through the coefficient of determination (R²), which was consistently above 0.98. All samples were tested in duplicates. Furthermore, the manufacturers state the sensitivity of the ELISA assays: 1 pg/mL for Aβ1-42 and 1.13 ng/L for γ-secretase. Due to these measures, we are confident that the ELISA results reflect accurate and precise quantification of the target proteins in both the hippocampal and frontal cortex homogenates. To assess the Reviewer’s concerns and to clarify our methodology we have added the above text to Section “Materials and Methods” 4.5, Lines 371-381.

Point #11

Considering the intricate nature of the development of Alzheimer's disease (AD), which involves various routes and variables, how accurately does the icvAβ1-42 + pin model reflect the sporadic type of AD that is commonly observed in humans?

Response: Thank you for this question as it supports the value of our model. Unlike familial AD, which occurs early due to mutations in the APP gene or its processing enzymes, sporadic AD lacks a single cause. A combination of chronic stress from environmental factors and/or lifestyle choices is thought to trigger sporadic AD. In most cases, the hallmark symptoms of AD arise not from excessive generation of Aβ1-42 but from the brain’s inability to efficiently clear the peptide. To model sporadic AD, several approaches have been developed, including intracerebroventricular (icv) injection of Aβ1-42 oligomers or fibrils. Other commonly used methods involve icv administration of agents such as streptozotocin, D-galactose, aluminum chloride, scopolamine, and colchicine (Khan, K.; et al. J. Explor. Res. Pharmacol. 2024, doi:10.14218/JERP.2023.00028). All of these interventions trigger inflammation, disturb the brain’s physiology and elicit AD-like symptoms. Moreover, AD is also characterized by melatonin deficiency, largely due to calcification and shrinkage of the pineal gland, which results in reduced melatonin levels and circadian rhythm disruption (Zhang, Z. et al. Mol. Psychiatry 2024 doi:10.1038/s41380-024-02691-6). In addition to pineal gland dysfunction, AD patients often experience a significant loss of intrinsically photosensitive retinal ganglion cells, which play a crucial role in regulating melatonin secretion (Mure, L.S. Front. Neurol. 2021, doi:10.3389/FNEUR.2021.636330/BIBTEX). As a result, many AD patients suffer from "sundowning," characterized by confusion and agitation in the late afternoon and evening, along with disturbances in their sleep-wake cycles and insomnia (Zhang, Z. et al. Mol. Psychiatry 2024 doi:10.1038/s41380-024-02691-6) (Lines 74-87). To mimic melatonin deficiency observed in AD patients and the corresponding symptoms, we combined icvAβ1-42 with pinealectomy. Given that these behavioral disturbances are closely linked to melatonin secretion by the pineal gland, we can confidently conclude that the icvAβ1-42 + pin model accurately represents a form of sporadic AD.

Point #12

Would the study be enhanced by include other behavioral tests to evaluate cognitive functions impacted by AD, such as anxiety or depressive-like behaviors, which are frequently observed in AD patients?

Response: Indeed, this information is extremely valuable to establish our model and support its accuracy in AD representation. This is why in our previous paper (doi: 10.3390/ijms222312763), where we first introduced the model, we validated the anxiety and depression of the animals. Specifically, we used the open field and elevated plus-maze tests. During the open field test, we monitored the total distance passed by the animals, the number of times they stand on their rear legs and the time they spend in the center. Pinealectomy, icvAβ1-42 administration and the combination of both significantly reduced the number of rears compared to the control, and showed a trend in reduced time spend in the center of the field. These observations confirmed that pin, icvAβ1-42 or both caused an anxiety response in the animals. The elevated plus maze test resulted in a similar behavior. The animals with pin, icvAβ1-42 or both entered much less the open arms of the maze and spend significantly less time in them compared to the control group. These results indicate that both pinealectomy and Aβ1-42 cause anxiety and depressive-like behaviors. Notably, melatonin improved their behavior, especially in the Aβ-pin group. In this manuscript we aimed to expand our knowledge on the icvAβ1-42 + pin rat model by testing their spatial memory ability. We have made a short note of that in Line 94, Line 231 and Lines 409-410, as we did not wish to confuse the readers that this is completely new information.

Point #13

The introduction asserts that sporadic AD generally originates from stress factors such as oxidative stress. However, the experimental emphasis appears to be predominantly on the metabolism of amyloid-beta. How do you harmonize these variables while formulating your opinions and suggestions for AD therapy?

Response: We made this statement based on our previous paper (doi: 10.3390/ijms222312763), where we tested the levels of total glutathione (GSH), superoxide dismutase (SOD) and lipid peroxidation via malondialdehyde (MDA). There we showed that pinealectomy reduced GSH, but increased SOD in both pin and icvAβ1-42 + pin groups. Additionally, lipid peroxidation was elevated in icvAβ1-42 ± pin. These results confirmed our hypothesis that Aβ1-42 increases oxidative stress in the hippocampus and frontal cortex, and that pinealectomy caused a disturbance in the balance in the antioxidant systems of SOD and GSH. Melatonin supplementation had little effect on SOD and GSH, but reduced MDA levels in the hippocampus, suggesting that melatonin acts as a direct free radical scavenger. Our previous study demonstrated that OS represents an essential part of brain pathology associated with pinealectomy and that the combination of icvAβ1-42 + pin causes more pronounced OS in the brain structures. Therefore, our previous and current results suggest that melatonin neuroprotection involves multiple targets and mechanism via either MT1/MT2 receptors or direct scavenging of free radicals. Considering that the administration of Aβ1-42 causes an increase in the levels of oxidative stress in model animals and the occurrence of cognitive changes, it can be concluded that the in vivo AD model is stress (oxidative)-induced and resembles the sporadic model of AD in humans. And to emphasize this fact we add the following phrase to Line 402: ” …further supported the oxidative stress-induced icvAβ1-42 + pin in vivo model……”, and on Lines 404-405: “Therefore, it represents the sporadic AD which affects the majority of AD patients...”

We would like to thank the Reviewer for the insightful comments, suggestions and ideas. We feel that addressing them has, without doubt, improved the overall quality of the manuscript and we are grateful for all of them.

Round 2

Reviewer 3 Report

Comments and Suggestions for Authors

The authors have replied to all my comments and questions satisfactorily.